# Effects of preservation methods of muscle tissue from upper-trophic level reef fishes on stable isotope values ($\delta^{13}$C and $\delta^{15}$N)

Christopher D. Stallings[1], James A. Nelson[2], Katherine L. Rozar[1], Charles S. Adams[1], Kara R. Wall[1], Theodore S. Switzer[3], Brent L. Winner[3] and David J. Hollander[1]

[1] College of Marine Science, University of South Florida, St. Petersburg, FL, USA
[2] Ecosystems Center, Marine Biological Laboratory, Woods Hole, MA, USA
[3] Florida Fish and Wildlife Conservation Commission, Fish and Wildlife Research Institute, St. Petersburg, FL, USA

Corresponding author
Christopher D. Stallings, stallings@usf.edu

## ABSTRACT

Research that uses stable isotope analysis often involves a delay between sample collection in the field and laboratory processing, therefore requiring preservation to prevent or reduce tissue degradation and associated isotopic compositions. Although there is a growing literature describing the effects of various preservation techniques, the results are often contextual, unpredictable and vary among taxa, suggesting the need to treat each species individually. We conducted a controlled experiment to test the effects of four preservation methods of muscle tissue from four species of upper trophic-level reef fish collected from the eastern Gulf of Mexico (Red Grouper *Epinephelus morio*, Gag *Mycteroperca microlepis*, Scamp *Mycteroperca phenax*, and Red Snapper *Lutjanus campechanus*). We used a paired design to measure the effects on isotopic values for carbon and nitrogen after storage using ice, 95% ethanol, and sodium chloride (table salt), against that in a liquid nitrogen control. Mean offsets for both $\delta^{13}$C and $\delta^{15}$N values from controls were lowest for samples preserved on ice, intermediate for those preserved with salt, and highest with ethanol. Within species, both salt and ethanol significantly enriched the $\delta^{15}$N values in nearly all comparisons. Ethanol also had strong effects on the $\delta^{13}$C values in all three groupers. Conversely, for samples preserved on ice, we did not detect a significant offset in either isotopic ratio for any of the focal species. Previous studies have addressed preservation-induced offsets in isotope values using a mass balance correction that accounts for changes in the isotope value to that in the C/N ratio. We tested the application of standard mass balance corrections for isotope values that were significantly affected by the preservation methods and found generally poor agreement between corrected and control values. The poor performance by the correction may have been due to preferential loss of lighter isotopes and corresponding low levels of mass loss with a substantial change in the isotope value of the sample. Regardless of mechanism, it was evident that accounting for offsets caused by different preservation methods was not possible using the standard correction. Caution is warranted when interpreting the results from specimens stored in either ethanol or salt, especially when using those from multiple preservation techniques. We suggest the use of ice as the preferred

preservation technique for muscle tissue when conducting stable isotope analysis as it is widely available, inexpensive, easy to transport and did not impart a significant offset in measured isotopic values. Our results provide additional evidence that preservation effects on stable isotope analysis can be highly contextual, thus requiring their effects to be measured and understood for each species and isotopic ratio of interest before addressing research questions.

# INTRODUCTION

The application of stable isotope analysis (SIA) has arguably been one of the most important innovations in the field of ecology in the last 50 years. SIA has been used across ecological sub-disciplines, providing a powerful tool to answer once intractable questions (*DeNiro & Epstein, 1981*; *Fry, 2006*; *Peterson & Fry, 1987*). Stable isotopes of carbon ($^{13}C/^{12}C$) and nitrogen ($^{15}N/^{14}N$) are innate components of all biological material, and the ratio of heavy to light isotopes observed in organisms is controlled by a confluence of biological and physical factors that fractionate the isotopes by differences in mass. These values are set by autotrophs and incorporated into the ecosystem as primary production is consumed (*O'Leary, 1988*). Carbon is typically used to identify primary production sources. For example, plants that use the C3 photosynthetic pathway have carbon isotope values depleted in the heavy isotope ($-28$ ‰) relative to grasses that use the C4 pathway ($-12$‰) (*O'Leary, 1988*). This difference has been used to determine when ancient cultures switched from gathering to farming (*Schoeninger & Moore, 1992*) and when brewers skirted Bavarian Purity Laws (*Brooks et al., 2002*). In contrast, nitrogen isotopes are often used to establish trophic position (*Post, 2002*). After food is consumed, metabolic processes preferentially cleave the bonds in proteins made with the lighter $^{14}N$ isotope. These waste products of metabolism are converted to urea and excreted leaving behind tissues made with the enriched $^{15}N$ amino acids (*Wright, 1995*). Typically, organisms are enriched approximately 3‰ relative to their food (*Hussey et al., 2014*; *Post, 2002*).

Research that uses stable isotope analysis often involves a delay between sample collection in the field and laboratory processing, therefore requiring preservation to prevent or reduce tissue degradation and associated changes in isotopic compositions. Methods used to preserve soft tissues, such as muscle, can present issues in the interpretation of the observed isotope values (*Sarakinos, Johnson & Zanden, 2002*). Although there is a growing literature describing the effects of various preservation techniques, the results are often unpredictable and vary among taxa, suggesting the need to treat each species individually (*Arrington & Winemiller, 2002*; *Correa, 2012*; *Kelly, Dempson & Power, 2006*; *Sarakinos, Johnson & Zanden, 2002*). When a systematic offset in isotope values is detected, a mass

**Table 1  Sample sizes, length information [mean (SE), minimum, and maximum], and C/N values of the focal species.**

| Species | No. collected | Mean (SE) TL (mm) | Max TL (mm) | Min TL (mm) | Mean (SE) C/N |
|---|---|---|---|---|---|
| *E. morio* | 24 | 569 (26) | 360 | 764 | 3.22 (0.13) |
| *M. microlepis* | 19 | 841 (38) | 500 | 1090 | 3.25 (0.15) |
| *M. phenax* | 15 | 569 (13) | 512 | 664 | 3.22 (0.15) |
| *L. campechanus* | 20 | 677 (16) | 546 | 794 | 3.22 (0.10) |

balance correction can be employed using the variation in C/N ratio to correct the isotope values of the preserved tissue (*Fry et al., 2003*; *Ventura & Jeppesen, 2009*). The underlying assumption of this method is that the preservation technique removes substances from the whole tissue (e.g., hydrolyzed lipids), altering the isotope value of the whole tissue, and this can be accounted for by relating the change in isotope value to the change in C/N ratio. However, these corrections are not always successful and there are still open questions about the mechanisms that alter tissue isotope values after preservation (*Kelly, Dempson & Power, 2006*).

In the current study, we conducted a controlled experiment to test the effects of four preservation methods of muscle tissue from four species of upper trophic-level reef fish (Red Grouper *Epinephelus morio*, Gag *Mycteroperca microlepis*, Scamp *Mycteroperca phenax*, and Red Snapper *Lutjanus campechanus*). We used a paired design to measure the effects on isotopic values for carbon and nitrogen after storage using ice, 95% ethanol, and sodium chloride (table salt), against that in a liquid nitrogen control. Additionally, we tested the application of standard mass balance corrections for isotope values that were significantly affected by the preservation methods.

## METHODS

### Collection and preservation of samples

Red Grouper, Gag, Scamp, and Red Snapper are co-occurring, essential members of reef ecosystems in the eastern Gulf of Mexico. They are ecologically important, mid- to upper-level predators that have also been among the most highly targeted fishes by commercial and recreational fishermen in the region.

Specimens were collected using hook-and-line from reef habitats in the eastern Gulf of Mexico as part of an ongoing fishery-independent study (Fig. 1). Collections of fishes were conducted in accordance with ethics policies followed by the University of South Florida Institutional Animal Care and Use Committee (approval no. W4193) and permits from the Florida Fish and Wildlife Conservation Commission (Special Activity License SAL-13-1244-SRP-2) and the US National Oceanic and Atmospheric Administration (Letter of Acknowledgment and Exempted Fishing Permit). A total of 78 individuals were collected for this study, across a range of sizes commonly observed for each species (Table 1). White muscle tissue ventral to the dorsal fin was removed from each specimen and cut into four, equal-sized pieces. Each piece was then subjected to one of four

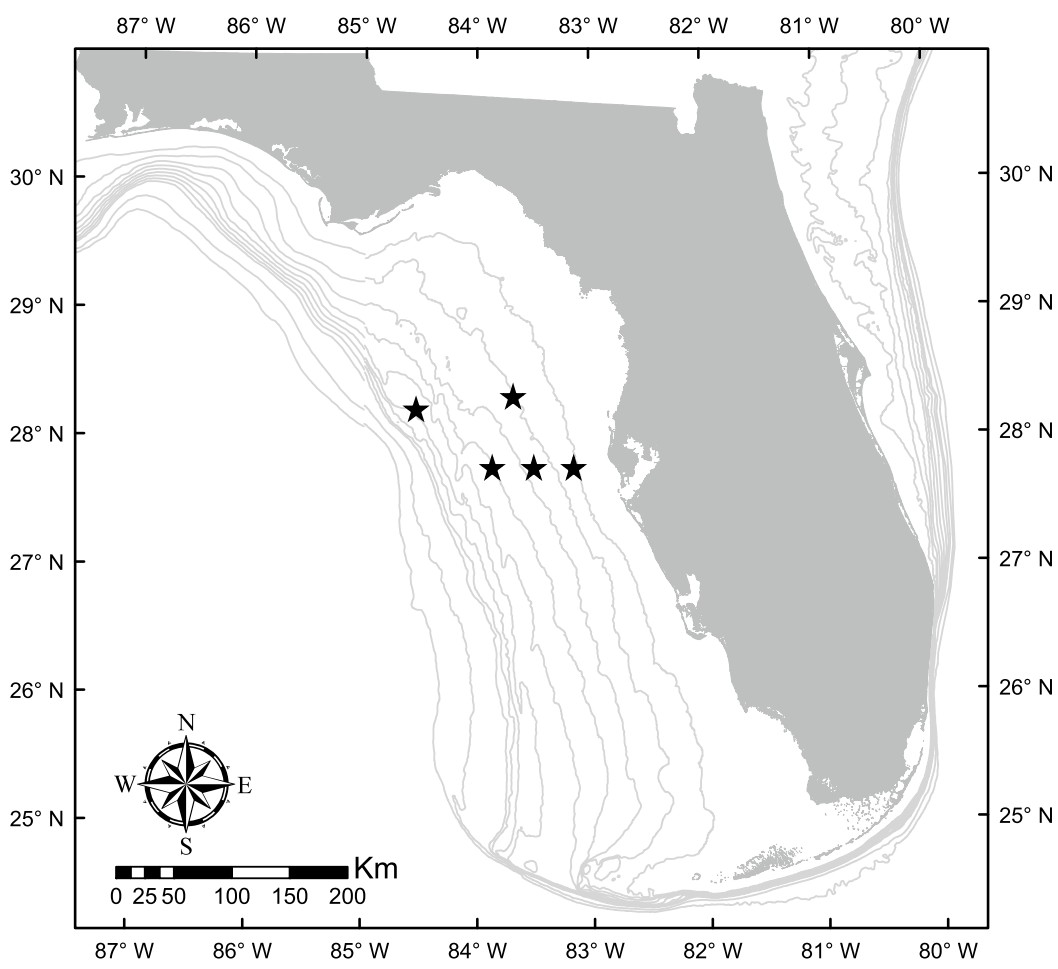

**Figure 1** Study region in the eastern Gulf of Mexico where samples were collected (locations of collection sites shown with black stars). 10 m isobaths are shown from 10–100 m.

preservation techniques flash freezing using liquid nitrogen (control), ice, 95% ethanol, or salt—all placed in uniquely-labeled, 2 ml microcentrifuge tubes. Control samples were frozen instantaneously by being placed in liquid nitrogen in a 4 liter vacuum flask. Liquid nitrogen served as a control as it neither effects existing isotopic values, nor does it allow bacterial degradation of the tissue to occur (*Michener & Lajtha, 2007*). Samples preserved with liquid nitrogen and those placed on ice were transferred to a −20 °C freezer after 48 h, representing a likely sequential scenario commonly used by field ecologists for tissue preservation. Freezing is one of the most commonly used controls for studies on preservation effects, as it has been shown to have negligible effects on isotope values of fish tissues (*Bosley & Wainright, 1999*; *Sarakinos, Johnson & Zanden, 2002*). For the other preservatives, samples were placed in microcentrifuge tubes with 1 ml of either 95% ethanol ($CH_3CH_2OH$) or table salt (NaCl), and were kept at ambient room temperature (22 °C). All samples were held for 30 days prior to processing for stable isotope analysis.

## Analytical procedures

At the conclusion of the preservation period, all tissues were rinsed with deionized water, and placed in glass vials in a drying oven (55–60 °C) for 48 h. Each desiccated muscle sample was then ground to a fine powder using a mortar and pestle to ensure even combustion during mass spectroscopy. The mortar and pestle, as well as additional tools and work surfaces, were cleaned with 99.5% ethanol and Kimwipes® between individual processing to prevent cross-contamination of samples. Ground samples with a dry weight of 200–1000 µg were placed in tin capsules and sealed for combustion and isotopic analysis. Using a Carlo-Erba NA2500 Series II elemental analyzer (Carlo Erba Reagents, Rodano, Milan, Italy) coupled to a continuous-flow ThermoFinnigan Delta + XL isotope ratio mass spectrometer, (Thermo Finnigan, San Jose, California, USA) we measured $^{13}C/^{12}C$, $^{15}N/^{14}N$ and C/N at the University of South Florida, College of Marine Science in St. Petersburg, Florida. The lower limit of quantification for this instrumentation was 12 µg C or N. We used calibration standards NIST 8573 and NIST 8574 L-glutamic acid standard reference materials. Analytical precision, obtained by replicate measurements of NIST 1577b bovine liver, was ±0.19‰ for $\delta^{15}N$ and ±0.11‰ for $\delta^{13}C$. Results are presented in standard notation ($\delta$, in ‰) relative to international standards Pee Dee Belemnite (PDB) and air.

## Mass balance corrections

We used an arithmetic correction based on changes in C/N and preserved vs control stable isotope values (*Fry et al., 2003*; *Smyntek et al., 2007*; *Ventura & Jeppesen, 2009*). This method assumes the preservation method alters the isotope values of the original tissue by leaching material into the preservative, specifically through the loss of hydrolyzed proteins or lipids. The assumption is that the loss of protein or lipid will be expressed by changes in the C/N of the preserved tissue and can be corrected by relating changes in isotope value of the preserved tissue to the change between the control and preserved tissue C/N as:

$$\delta_{control} = \delta_{preserved} - \Delta\delta_{(preserved-control)} \tag{1}$$

$$\Delta\delta_{(preserved-control)} = X\left(\frac{C/N_{control} - C/N_{preserved}}{C/N_{preserved}}\right) \tag{2}$$

where the $\delta_{control}$ is the isotope value of the unpreserved tissue and $\delta_{preserved}$ is the isotope value of the preserved tissue. $\Delta\delta_{(preserved-control)}$ is the net effect of preservation of the isotope value of the preserved tissue. $X$ is the difference between the isotope value of the preserved and control tissue.

## Statistical analysis

We provide mean (SE) offset values for preservative—control both across and within species. For each species, we used paired *t*-tests to determine whether $\delta^{15}N$ and $\delta^{13}C$ isotopic values from preserving samples with ice, ethanol, and salt were statistically different from control samples preserved in liquid nitrogen. We also use linear regression with 95% confidence intervals of corrected against control isotopic values to determine efficacy of the mass balance corrections.

**Table 2** Mean (SE) offset (‰) for nitrogen and carbon isotopes across four focal species based on absolute values of preservative–control.

| Preservative | $\delta^{15}N$ | $\delta^{13}C$ |
|---|---|---|
| Ice | 0.20 (0.02) | 0.28 (0.03) |
| EtOH | 0.56 (0.04) | 0.42 (0.04) |
| NaCl | 0.47 (0.04) | 0.34 (0.05) |

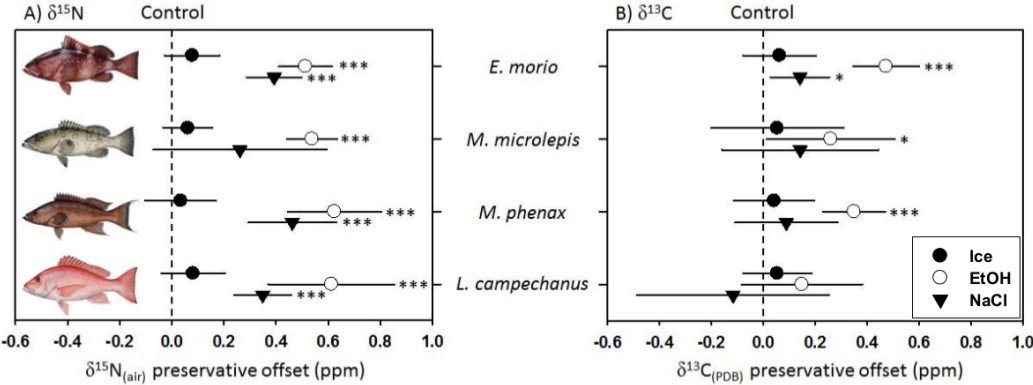

**Figure 2** Offsets (preservative–control) in (A) $\delta^{15}N$ and (B) $\delta^{13}C$ isotopic values due to preservation technique (mean ± 2 SE). Offsets for preservatives that were statistically different from the liquid nitrogen controls are noted as $*(P < 0.05)$, $**(P < 0.01)$, and $***(P < 0.001)$. Fish illustrations courtesy of Diane Peebles.

## RESULTS

Across species, mean offsets for both $\delta^{13}C$ and $\delta^{15}N$ values from controls were lowest for samples preserved on ice, and highest for those preserved with ethanol (Table 2). Offsets for $\delta^{15}N$ were generally higher than those for $\delta^{13}C$. For $\delta^{13}C$, salt imparted a 21% offset and ethanol a 50% offset compared to ice. For $\delta^{15}N$, salt imparted a 135% offset compared to ice and ethanol a 180% offset.

Within species, the effects of the different preservatives ranged in both magnitude and statistical significance (Fig. 2 and Table 3). Ethanol preservation significantly affected $\delta^{15}N$ values in all four species, and $\delta^{13}C$ values in all three groupers. Salt preservation significantly affected $\delta^{15}N$ values in three species (Red Grouper, Scamp and Red Snapper), and $\delta^{13}C$ in only Red Grouper. Ice preservation did not impart a strong or statistically significant offset in either isotope ratio of any species measured.

### C/N corrections

Overall, there was poor agreement between corrected and control values using the mass balance approach. Although all regressions were within 95% confidence of a 1:1 slope (i.e., slope = 1, intercept = 0), with the exception of ethanol-preserved $\delta^{13}C$ for Red Snapper, the fit was low for both corrected ethanol- (mean $R^2 = 0.23$; $R^2$ range $= < 0.01$–0.39; Figs. 3–6A and 6B) and salt-treated samples (mean $R^2 = 0.23$; range $= < 0.01$–0.55; Figs. 3–6C and 6D) across all four species. Corrected values for both preservatives fell

**Table 3 Summary of paired *t*-tests and two-sided *P*-values across species and preservation method for $\delta^{15}N$ and $\delta^{13}C$ isotopes. Significant *P*-values are bold-typed.**

| Species | Preservation(df) | $\delta^{15}N$ | | $\delta^{13}C$ | |
|---|---|---|---|---|---|
| | | *t*-value | *P*-value | *t*-value | *P*-value |
| *E. morio* | | | | | |
| | ice(23) | −1.494 | 0.149 | −0.885 | 0.385 |
| | ethanol (22) | −9.956 | **<0.001** | −7.446 | **<0.001** |
| | salt(23) | −7.400 | **<0.001** | −2.472 | **0.021** |
| *M. microlepis* | | | | | |
| | ice(18) | −1.276 | 0.218 | −0.418 | 0.681 |
| | ethanol(18) | −11.077 | **<0.001** | 2.438 | **0.025** |
| | salt(15/18) | −1.569 | 0.134 | −0.939 | 0.360 |
| *M. phenax* | | | | | |
| | ice(13) | −0.498 | 0.627 | −0.533 | 0.603 |
| | ethanol(12) | −6.906 | **<0.001** | −5.794 | **<0.001** |
| | salt(13) | −5.464 | **<0.001** | −0.892 | 0.389 |
| *L. campechanus* | | | | | |
| | ice(19) | −1.318 | 0.203 | −0.802 | 0.432 |
| | ethanol(18) | −5.040 | **<0.001** | −1.269 | 0.221 |
| | salt(18) | −6.246 | **<0.001** | −0.189 | 0.852 |

**Table 4 Mean (SE) change in C/N due to ethanol and salt preservation methods.**

| Species | EtOH | NaCl |
|---|---|---|
| *E. morio* | 0.06 (0.19) | 0.03 (0.15) |
| *M. microlepis* | 0.02 (0.20) | −0.18 (0.57) |
| *M. phenax* | 0.04 (0.20) | −0.02 (0.14) |
| *L. campechanus* | −0.09 (0.22) | −0.07 (0.24) |

on both sides of the 1:1 line, thus our correction did not tend to systematically under or overestimate the change in nitrogen isotope values after preservation. The poor correction values were a direct consequence of the small change and small degree of correlation between the change in the C/N ratio of the control and preserved tissues relative to the change in isotope values (Table 4).

## DISCUSSION

Using a controlled experiment, we have demonstrated that three techniques used to preserve muscle tissue can have varying effects on measured isotope values for four species of reef fish. Both ethanol and salt caused significant changes to the measured isotope values, but the effects were contextual on species and the isotope being measured. Conversely, preservation of muscle tissue on ice for 48 h, followed by storage in a −20 °C freezer for 28 days, did not impart a significant offset in the isotopic values of either carbon or nitrogen for any of our focal species. Because ice is widely available, inexpensive, and easy to transport relative to liquid nitrogen, we suggest its use as a preservation technique

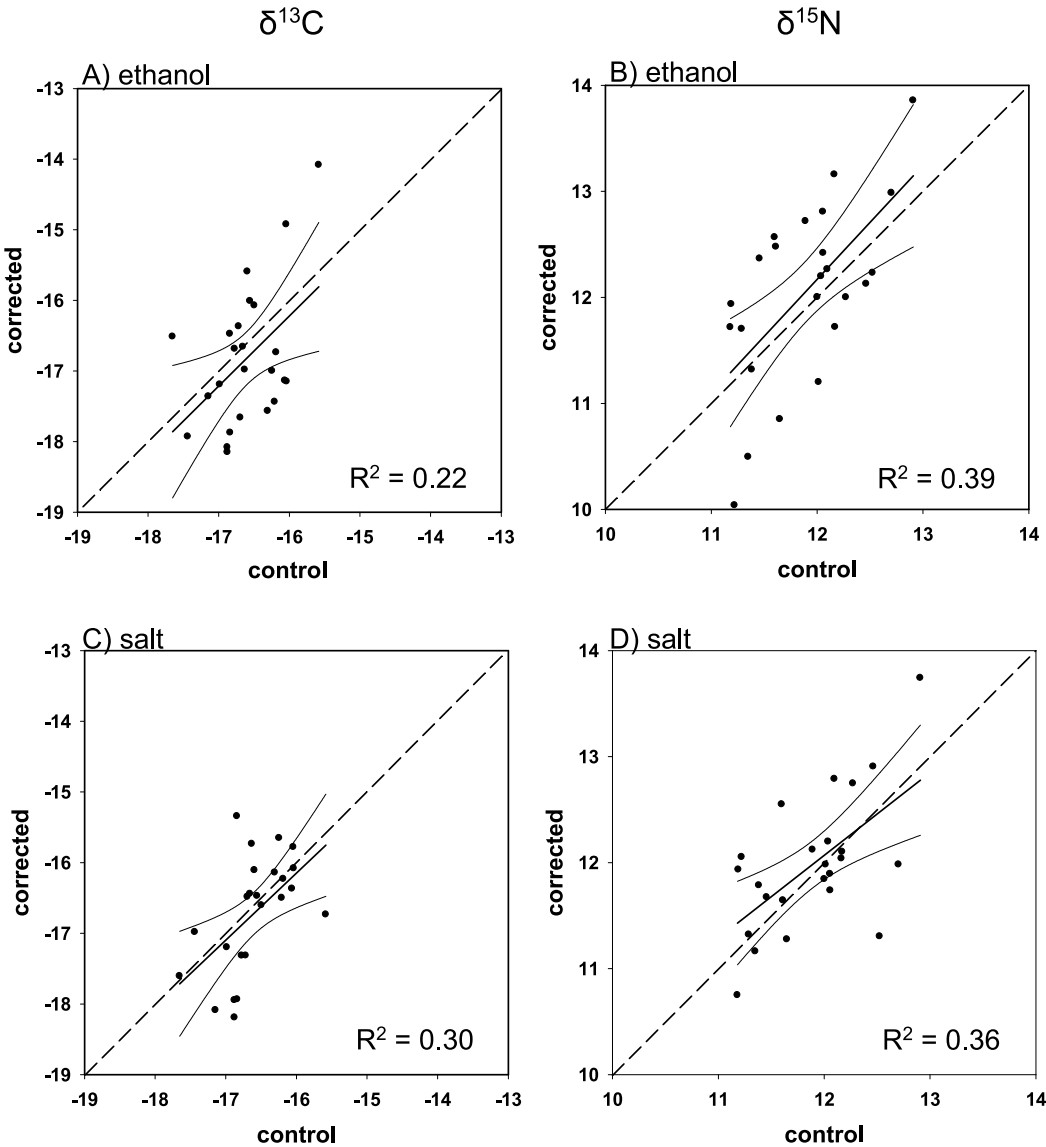

**Figure 3 Mass balance corrected values against control values for *E. morio*.** The 1:1 line (hashed), predicted (solid), and 95% confidence intervals are given for (A) ethanol $\delta^{13}$C, (B) ethanol $\delta^{15}$N, (C) salt $\delta^{13}$C, and (D) salt $\delta^{15}$N.

for muscle tissue from Red Grouper, Gag, Scamp and Red Snapper when conducting stable isotope analysis.

There is a substantial and growing number of studies on the effects of various preservatives and methods on carbon and nitrogen stable isotope values in animal tissues (*Barrow, Bjorndal & Reich, 2008*; *Sarakinos, Johnson & Zanden, 2002*; *Ventura & Jeppesen, 2009*). Despite the large body of work on the topic, there is little consensus on the effect of preservation techniques on stable isotope values with a near even number of studies finding significant and non-significant shifts (*Kelly, Dempson & Power, 2006*; *Sweeting, Polunin & Jennings, 2004*; *Ventura & Jeppesen, 2009*). When significant differences between

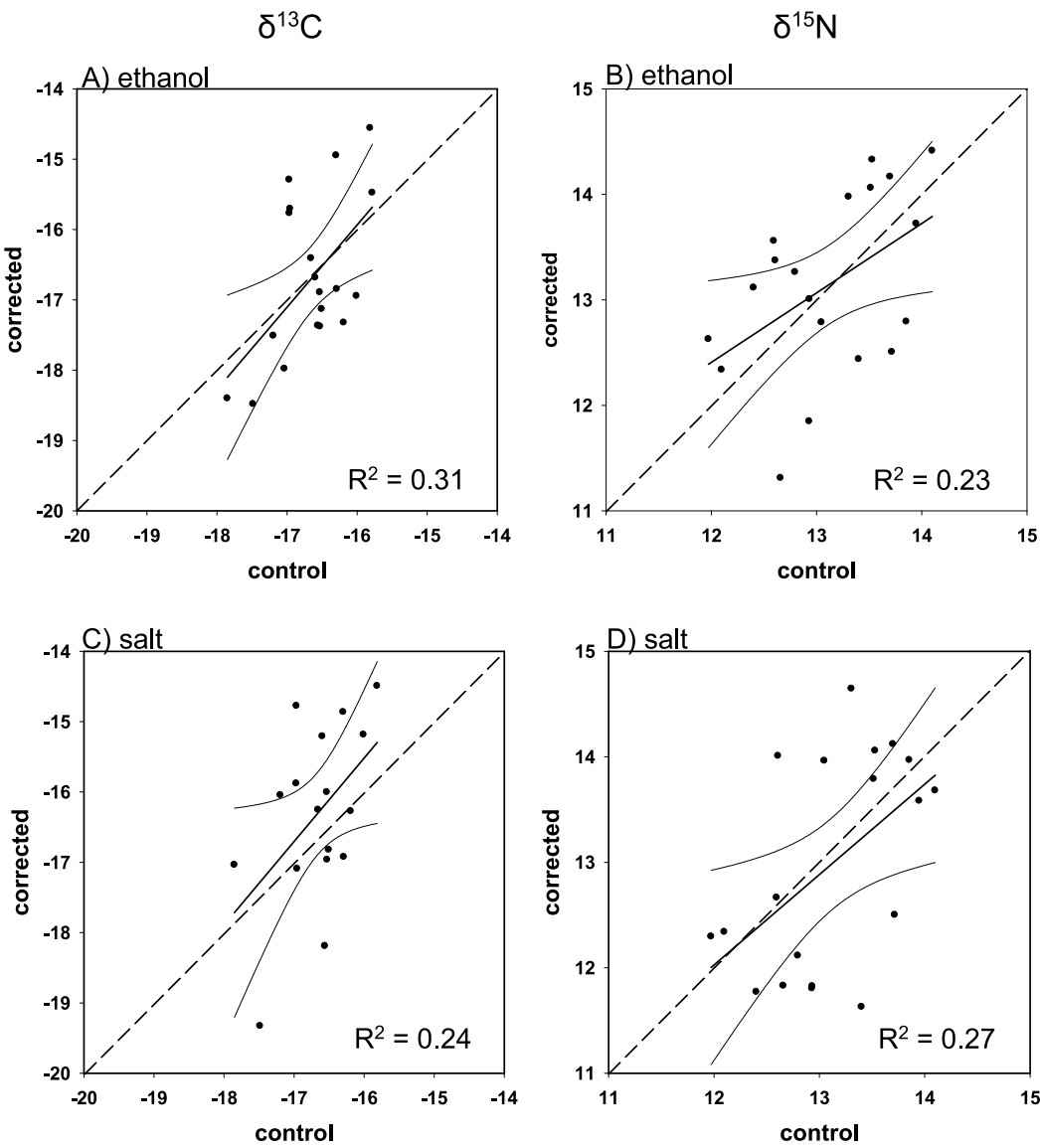

**Figure 4 Mass balance corrected values against control values for *M. microlepis*.** The 1:1 line (hashed), predicted (solid), and 95% confidence intervals are given for (A) ethanol $\delta^{13}$C, (B) ethanol $\delta^{15}$N, (C) salt $\delta^{13}$C, and (D) salt $\delta^{15}$N.

control and preserved tissues have been observed, researchers most often opt to develop a correction curve based on the variation in the C/N in the preserved tissues (*Fry et al., 2003*; *Logan et al., 2008*; *Sarakinos, Johnson & Zanden, 2002*). However, our results show that even very small changes in C/N can co-occur with a slight enrichment in carbon and significant enrichment in nitrogen stable isotope values (Fig. 2).

For carbon isotope values, the enrichment was statistically significant in 3 of the 4 fishes examined for ethanol preservation. We conclude this slight enrichment was caused by the loss of lipids from the tissue. Lipids are depleted in $^{13}$C relative to the sugars they are created from by approximately 7‰ as a result of fractionation during the

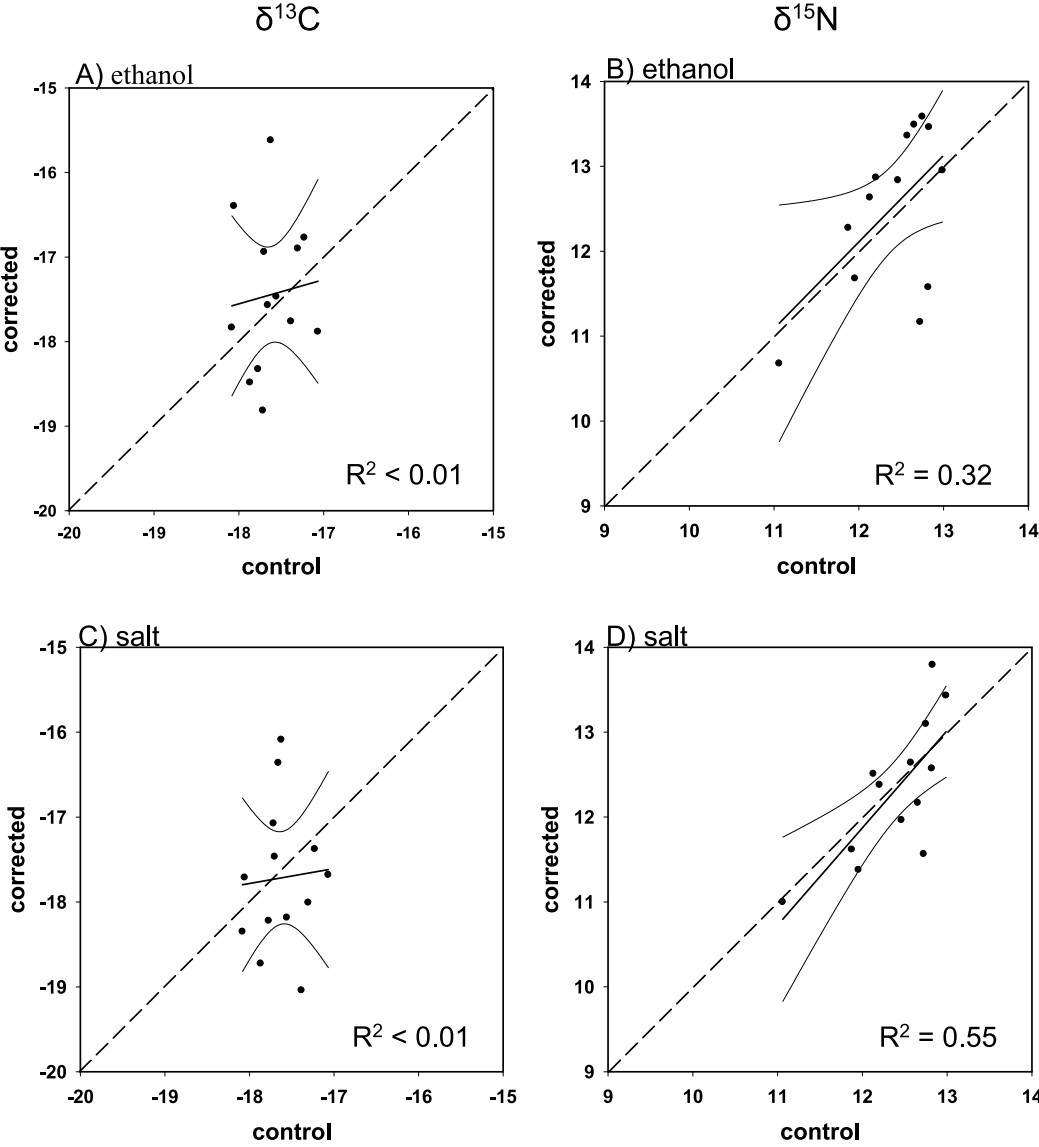

**Figure 5  Mass balance corrected values against control values for *M. phenax*.** The 1:1 line (hashed), predicted (solid), and 95% confidence intervals are given for (A) ethanol $\delta^{13}$C, (B) ethanol $\delta^{15}$N, (C) salt $\delta^{13}$C, and (D) salt $\delta^{15}$N.

oxidation of pyruvate to acetyl coenzyme A (*DeNiro & Epstein, 1977*). Ethanol is relatively non-polar compared to the water in the muscle tissue and could extract the lipids into the preservative, and indeed enrichment of tissues has been shown after the application of lipid extraction techniques to muscle tissue (*Logan et al., 2008*; *Nelson et al., 2011*). All of the pre-extraction tissues had C/N ratios typical of fish muscle, ∼3.4, prior to preservation and showed slight changes in isotope values typical of those observed in previous studies (*Logan et al., 2008*; *Nelson et al., 2011*).

All fishes showed a significant enrichment of nitrogen isotope value with ethanol and salt preservation with little change in C/N ratio. This indicates there was a significant loss of

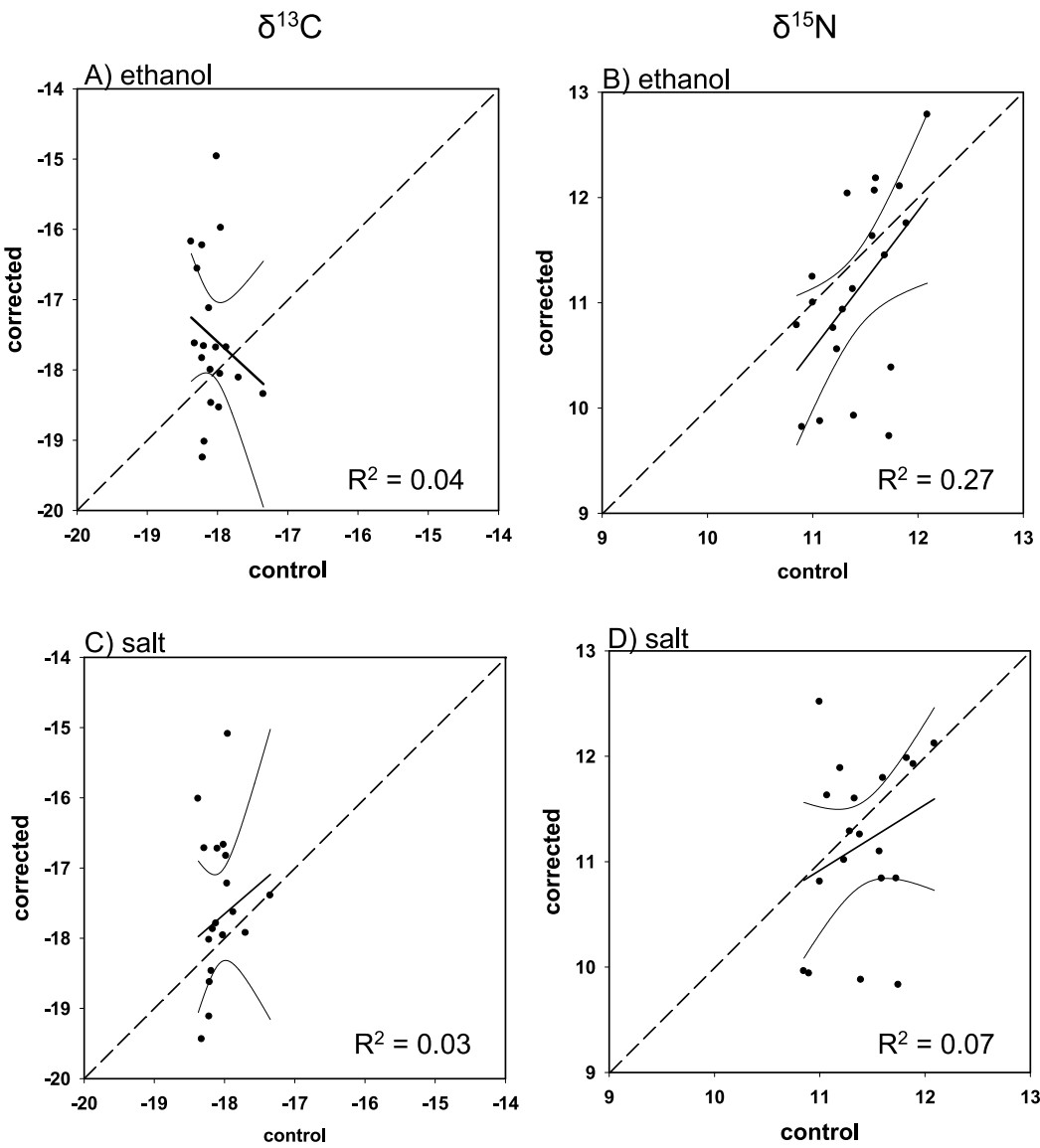

**Figure 6** **Mass balance corrected values against control values for *L. campechanus*.** The 1:1 line (hashed), predicted (solid), and 95% confidence intervals are given for (A) ethanol $\delta^{13}C$, (B) ethanol $\delta^{15}N$, (C) salt $\delta^{13}C$, and (D) salt $\delta^{15}N$.

the light isotope from tissue upon preservation with little change in mass from the sample itself. Therefore, we conclude there were some critical fractionation processes associated with the loss of amino acids linked with the breakdown of proteins. This resulted in low levels of mass loss but a substantial change in the isotope value of the sample.

There are a two mechanisms that may be responsible for the observed results. Ethanol is known to denature proteins and form new bonds between ethanol and the protein side chains (*Herskovits, Gadegbeku & Jaillet, 1970*; *Nozaki & Tanford, 1971*). The free energy required to conduct these reactions is high and therefore likely favors the cleaving of $^{14}N$–$^{14}N$ bonds. Thus, such reactions may explain the very high fractionation yet low

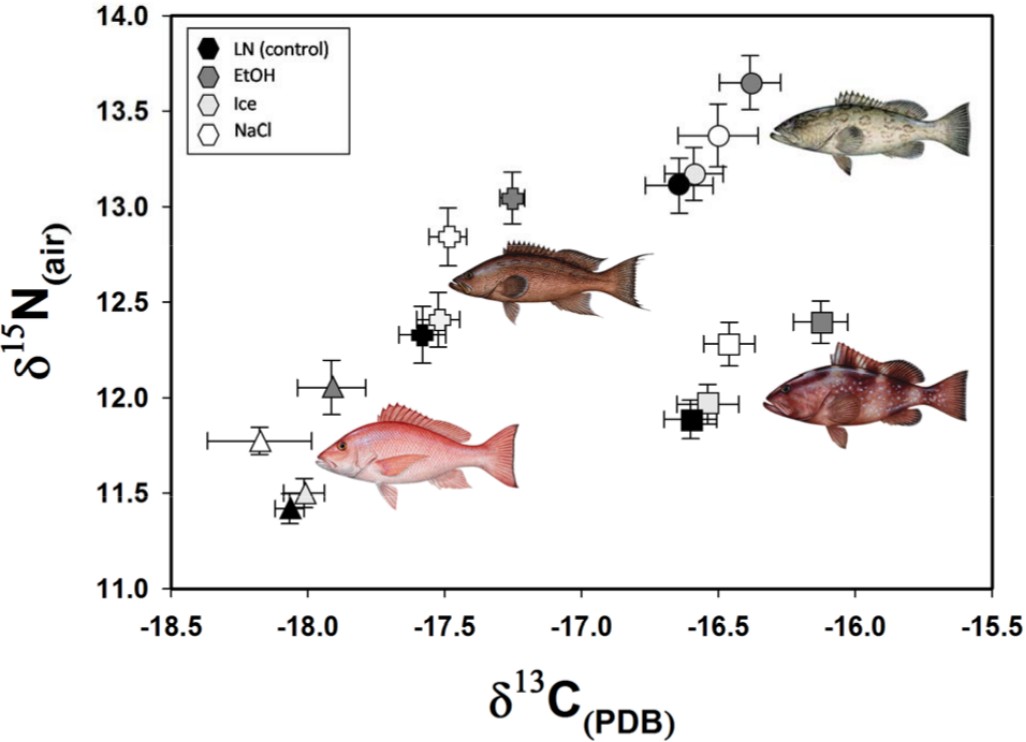

**Figure 7** Carbon and nitrogen isotope values for the four study species showing the relative trophic positions and the effects of different preservation methods. Fish illustrations courtesy of Diane Peebles.

mass loss observed in this study with preservation in ethanol. We also observed a strong fractionation of the nitrogen isotope values of the preserved tissues with salt. Salt is highly effective at extracting proteins from tissue samples, removing as much as 91% of the available protein (*Dyer, French & Snow, 1950*). Because our samples were stored at a low temperature, the extraction was likely less efficient. Regardless, it preferentially removed the light nitrogen bonds, resulting in little mass loss with high fractionation.

We designed our experiment to represent a "typical" sampling, preservation, and processing time used in ecological studies. The rates of the processes described above would all vary with changes in time, temperature, preservative volume, and physical dimensions of the sample (e.g., surface area to volume ratio). In addition, differences among species in the protein and lipid content of the muscle tissue could affect the post-preservation isotope values. Exposure to preservatives longer than 30 days or samples with higher lipid contents may produce greater changes in isotope values after preservation. It is our suggestion that given both preservatives are known to extract proteins, amino acids, and lipids with the potential for an unknown amount of fractionation to occur in proteins, caution be used when interpreting the results from specimens stored in either ethanol or salt.

To further illustrate why caution is warranted for interpreting isotope values for specimens preserved in ethanol or salt, we provide a standard biplot with mean (SE) values of $\delta^{13}C$ and $\delta^{15}N$ for each species-by-preservation method (Fig. 7). In

$\delta^{13}$C–$\delta^{15}$N space, samples preserved in liquid nitrogen and on ice are indistinguishable from each other. However, a strong departure from control values is evident especially in $\delta^{15}$N space for salt and even more so for ethanol. While the ecological importance of these statistically significant offsets would be dependent upon the questions being asked, the biplot illustrates how both quantitative and qualitative conclusions may be hindered by ethanol and salt preservation. This observation could be further exacerbated by comparing stable isotope values from samples preserved with multiple techniques in the same study. For example, mixing isotopic values from tissues preserved differently could lead to misleading conclusions regarding niche space (e.g., *Layman et al., 2007*). Researchers using stable isotope analysis on the species presented here, and any others involving soft tissues, should either use a common preservation method, or at a minimum, understand the potential effects of different preservation methods before making cross-study comparisons. Further work should also be conducted to determine whether long-term storage, including freezing, may have important effects on isotope values before historic specimens (e.g., museum collections) are used. Our results provide additional evidence that preservation effects on stable isotope analysis can be highly contextual, thus requiring their effects to be measured and understood for each species and isotopic ratio of interest before addressing research questions.

## ACKNOWLEDGEMENTS

We thank the boat owners, captains and mates that participated in our field research, including Bandit Charters (F/V Lady Rose—Captain Tom Rice), Charisma Charters (F/V Charisma—Captain Charles "Chuck" Guilford), F/V Offshore Account (Captain Matt Tevlin) and Get Reel Fisheries (F/V The Patriot—Captain Tan Beal, owner Reba Ferrell). This work involved collaboration with the Gulf of Mexico Reef Fish Shareholders' Alliance, especially with TJ Tate, David Krebs, and Jason Delacruz. We gratefully acknowledge the field staff of the Fish and Wildlife Research Institute that put in countless hours collecting and processing project-associated data, especially Caleb Purtlebaugh. Oversight of the isotope processing was provided by Ethan Goddard and brute musings regarding protein solutions by Vic Chessnut.

### Funding

Funding was provided by a grant to CD Stallings and TS Switzer from the National Oceanic and Atmospheric Administration, Cooperative Research Program (NA12NMF4540081). The funders had no role in study design, data collection and analysis, decision to publish, or preparation of the manuscript.

### Grant Disclosures

The following grant information was disclosed by the authors:
National Oceanic and Atmospheric Administration, Cooperative Research Program: NA12NMF4540081.

## Competing Interests

James A. Nelson is an employee of the Marine Biological Laboratory at Woods Hole, and both Theodore S. Switzer and Brent L. Winner are employees of the Florida Fish and Wildlife Conservation Commission, Fish and Wildlife Research Institute.

## Author Contributions

- Christopher D. Stallings conceived and designed the experiments, performed the experiments, analyzed the data, wrote the paper, prepared figures and/or tables, reviewed drafts of the paper.
- James A. Nelson analyzed the data, wrote the paper, prepared figures and/or tables, reviewed drafts of the paper.
- Katherine L. Rozar performed the experiments, analyzed the data, wrote the paper, prepared figures and/or tables, reviewed drafts of the paper.
- Charles S. Adams performed the experiments.
- Kara R. Wall performed the experiments, contributed reagents/materials/analysis tools, reviewed drafts of the paper.
- Theodore S. Switzer, Brent L. Winner and David J. Hollander contributed reagents/materials/analysis tools.

## Animal Ethics

The following information was supplied relating to ethical approvals (i.e., approving body and any reference numbers):

University of South Florida Institutional Animal Care and Use Committee (approval no. W4193).

## Field Study Permissions

The following information was supplied relating to field study approvals (i.e., approving body and any reference numbers):

Collections of fishes were conducted with permits from the Florida Fish and Wildlife Conservation Commission (Special Activity License SAL-13-1244-SRP-2) and the US National Oceanic and Atmospheric Administration (Letter of Acknowledgment and Exempted Fishing Permit).

## Supplemental Information

Supplemental information for this article can be found online at http://dx.doi.org/10.7717/peerj.874#supplemental-information.

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
