# Peer review of "Effects of preservation methods of muscle tissue from upper-trophic level reef fishes on stable isotope values (δ13C and δ15N)"

_PeerJ, doi:10.7717/peerj.874_

## Round 0.1 · original submission · Minor Revisions

· Academic Editor

Minor Revisions

The MS has been reviewed by two independent referees who both find the study and the findings of interest but raise a few questions. The authors are requested to address these issues.

Reviewer 1 ·

Basic reporting

In general, this manuscript is well-written and clear. The authors have assumed that protein or lipids are being extracted by preservative, and in general have a good discussion about possibility of protein extraction. I feel they are missing some key references and discussion of the effects of lipid extraction on C:N ratios (further discussed in "Validity of the Findings"), and including this may improve their manuscript. I particularly encourage them to review Logan, J. M., Jardine, T. D., Miller, T. J., Bunn, S. E., Cunjak, R. A., & Lutcavage, M. E. (2008). Lipid corrections in carbon and nitrogen stable isotope analyses: comparison of chemical extraction and modelling methods. Journal of Animal Ecology, 77(4), 838-846. I am not convinced that Tables 2 and 3 are necessary. Table 2 could easily be incorporated into the text on page 8 (line 114) of the submission and Table 3 appears to duplicate information that is better illustrated in Figure 2. Figure 1 would benefit from additional location information (i.e. were the fish collected from anywhere within the dashed box?). The caption for Figure 2 could benefit from additional description of what "A" and "B" depict, plus what the dashed line denotes. The captions for Figures 3-6 could benefit from additional reminders of what "corrected" means. The figures themselves could also benefit from clearer demarcation of which panels depict ethanol or salt (i.e. labels right on the figure, as with the d15N and d13C labels, rather than in the caption). Figure 7 is interesting but I am not sure the results are quite as alarming as the authors seem to suggest (further discussed in "Validity of the Findings"). There are minor issues with sentence flow in the methods section, and a typo on page 7 line 85 (Carlo-Erba). I also suggest that the authors maintain consistent use of either scientific or common names throughout the figures.

Experimental design

The research question is clearly defined, relevant and meaningful given that most of the literature suggests species-level correction factors are needed. There may be a question as to whether the "control" is actually a control since it was also preserved for 30 days rather than analyzed straight away. I believe there are studies that suggest flash-freezing fish muscle tissue results in negligible impacts on the carbon and nitrogen stable isotope signatures of said tissue, and the manuscript would benefit from discussion of this. It would be more appropriate to use one ANOVA per species (comparing between all preservation methods) rather than multiple t-tests. If the authors have the ability (i.e. data collected but not presented and/or specimens available to be analyzed), the manuscript may benefit from inclusion of bulk C:N values ("unpreserved"), bulk d13C values and lipid-free d13C (i.e. not just preserved and assumed to be lipid or protein-depleted) values for each species.

Validity of the findings

The main issue I see is a lack of discussion of effects of lipid extraction on d13C and d15N values. The authors could do a better job of acknowledging that preservation does not only affect protein in tissues (which they do discuss), but is well-known to affect lipids. The manuscript could benefit from a discussion of the expected rates of exchange of protein and lipid between the preserved tissue and the preservative (i.e. is 30 days long enough to see changes in all species for all preservatives? Is it possible that exchange rates for some species would be slower than others? Is it possible that not all of the lipids were extracted and this may be why the C:N correction does not appear to be adequate for some species?). The results depicted in Figures 3-6 should be considered in this light. Figure 7 is interesting and I wonder if discussion of the relative fat content of each species would help explain some of the magnitude of differences (I am unfamiliar with these species). At the same time, given the relatively small differences in terms of things like trophic level, I am not sure that the strong language used in the discussion in relation to Figure 7 is warranted. Suggesting caution is fine. The final piece of the discussion where the authors attempt to connect to the Deep Horizon spill is somewhat awkward and out of place. I suggest the authors include more discussion of the possible applications of these types of studies to broader issues (because there are many - possibilities to consider are the use of historic specimens preserved in ethanol being compared to contemporary specimens and the ease of use/reliability/suitability of salt preservation vs. freezing - I believe other studies have suggested that salt is a reliable alternative but this study seems to contradict that), if they wish to include this piece. This may require slight edits to the introduction to ensure the discussion pieces do not seem out of place.

Reviewer 2 ·

Basic reporting

No comments.They are included in the General Comment Section

Experimental design

No comments.They are included in the General Comment Section

Validity of the findings

No comments.They are included in the General Comment Section

Additional comments

The paper examines the effects of preservation methods of muscle tissue on stable isotope values. Its value is of practical importance as the application of SIA is a relatively recent innovation in ecology studies and its application is spreading nowadays so it worthwhile to be published.
The objective of the paper are clearly stated and the conclusions drawn consequently. Clear and unambiguous text has been used.
While the authors suggest ready explanations for the differences observed according to three preservation techniques, the significant changes registered among the considered fish species have not been discussed or even speculated. I suggest to reinforce the paper with some considerations in this respect.

---

## Round 0.2 · accepted · Accept

· Academic Editor

Accept

We appreciate that you have given due consideration to the comments and questions raised by the two referees. We also noted that appropriate changes have been made in the MS, where explanations / changes / additions were required.